# Sensory Integration: A Novel Approach for Healthy Ageing and Dementia Management

**DOI:** 10.3390/brainsci14030285

**Published:** 2024-03-18

**Authors:** Ongart Maneemai, Maira Cristina Cujilan Alvarado, Lina Graciela Calderon Intriago, Alicia Jeanette Donoso Triviño, Joicy Anabel Franco Coffré, Domenico Pratico, Kristof Schwartz, Tadele Tesfaye, Takao Yamasaki

**Affiliations:** 1Department of Pharmaceutical Care, School of Pharmaceutical Sciences, University of Phayao, Phayao 56000, Thailand; ongart.ma@up.ac.th; 2School of Nursing, University of Guayaquil, Guayaquil 090510, Ecuador; maira.cujilana@ug.edu.ec (M.C.C.A.); lina.calderoni@ug.edu.ec (L.G.C.I.); alicia.donosot@ug.edu.ec (A.J.D.T.); joicy.francoc@ug.edu.ec (J.A.F.C.); 3Alzheimer’s Center at Temple, Lewis Katz School of Medicine, Temple University, Philadelphia, PA 19140, USA; 4Institute of Health Promotion and Sport Sciences, ELTE Eötvös Loránd University, 1053 Budapest, Hungary; 5CareHealth Medical Practice, Jimma Road, Addis Ababa 9023, Ethiopia; 6Department of Neurology, Minkodo Minohara Hospital, Fukuoka 811-2402, Japan; 7Kumagai Institute of Health Policy, Fukuoka 816-0812, Japan

**Keywords:** sensory integration, dementia, neuromodulation, healthy aging

## Abstract

Sensory processing is a fundamental aspect of the nervous system that plays a pivotal role in the cognitive decline observed in older individuals with dementia. The “sensory diet”, derived from sensory integration theory, may provide a tailored approach to modulating sensory experiences and triggering neuroplastic changes in the brain in individuals with dementia. Therefore, this review aimed to investigate the current knowledge regarding the sensory diet and its potential application to dementia. This review encompassed an extensive search across multiple databases, including PubMed, Google Scholar, covering articles published from 2010 to 2023. Keywords such as “sensory integration”, “sensory modulation”, “healthy aging”, and “dementia” were utilized to identify relevant studies. The types of materials retrieved included peer-reviewed articles, systematic reviews, and meta-analyses, ensuring a comprehensive overview of the current research landscape. This article offers a comprehensive exploration of the effectiveness of sensory diets such as tactile stimulation, auditory therapies, and visual interventions, which have demonstrated noteworthy efficacy in addressing challenges linked to aging and dementia. Research findings consistently report positive outcomes, such as improved cognitive function, elevated emotional well-being, and enhanced overall quality of life in older individuals. Furthermore, we found that the integration of sensory diets with the metaverse, augmented reality, and virtual reality opens up personalized experiences, fostering cognitive stimulation and emotional well-being for individuals during aging. Therefore, we conclude that customized sensory diets, based on interdisciplinary cooperation and leveraging technological advancements, are effective in optimizing sensory processing and improve the overall well-being of older individuals contending with sensory modulation challenges and dementia.

## 1. Introduction

Sensory processing is a crucial function of the nervous system, and impairments in sensory processing significantly affect a large portion of the population, particularly older individuals who may be dealing with different forms of dementia. The global population of people aged 65 or older has reached a substantial count of around 703 million, representing approximately 9% of the total global population. Within this demographic, an estimated 5–8% experience sensory dysfunction, highlighting the growing need to address sensory modulation challenges for healthy aging [1,2].

The notion of a “sensory diet” represents a novel and emerging paradigm aimed at optimizing sensory processing mechanisms in individuals, with implications for various neurobehavioral contexts. Rooted in the foundational principles of sensory integration theory, sensory diets are intricately designed interventions predicated on the precise manipulation of sensory stimuli and the natural progression that individuals undergo as they age and involve activities to ameliorate sensory processing anomalies [3,4]. Fundamentally, the human nervous system is inherently responsible for receiving, interpreting, and responding to sensory inputs from the environment. Dysfunctions in this process can manifest as sensory sensitivities, hyposensitivity, or sensory-seeking behaviors, impacting an individual’s daily functioning [5]. Sensory diets, based on the theoretical framework by A. Jean Ayres, have gained scientific attention for their potential to address these issues [6]. These diets encompass a diverse range of sensory modalities, including tactile, proprioceptive, vestibular, auditory, and visual stimuli, methodically integrated into personalized regimens [7].

In recent times, advancements in the field of sensory diets have been significant, driven by ongoing research and an increasing understanding of sensory processing. These advancements aim to optimize the effectiveness of sensory interventions for individuals with sensory modulation difficulties. Several key areas where advancements have occurred include:Personalization and Precision: A notable progression in the field of sensory diets involves an increased emphasis on personalized interventions. Rather than adhering to a one-size-fits-all methodology, contemporary experts recognize the pivotal importance of tailoring sensory interventions to an individual’s unique sensory profile. This involves comprehensive assessments and the integration of an individual’s sensory preferences and sensitivities into the intervention plan [8].Technology Integration: Technological advancements, particularly in virtual reality (VR), sensory applications, and augmented reality (AR) have played a pivotal role in advancing sensory diets. These innovations incorporated sensory interventions, providing immersive and controlled sensory experiences in therapeutic settings. Therapists utilize these technologies to create customized sensory inputs, optimizing individualized interventions. Moreover, the integration of VR, sensory apps, and AR enhanced the precision and efficacy of patient progress monitoring, marking a transformative shift towards sophisticated and tailored sensory interventions [9].Tele-health and Remote Interventions: Telehealth applications have emerged as valuable tools for delivering interventions for dementia, providing an array of benefits including enhanced accessibility and cost-effectiveness. Remote monitoring features enable the continuous assessment of cognitive functions through cognitive tests and the tracking of daily activities using wearable devices and sensors. Telehealth assessments, facilitated through video conferencing, allow regular consultations with healthcare professionals, facilitating adjustments to treatment plans and addressing the well-being of individuals with dementia. Moreover, cognitive training and rehabilitation programs delivered through online platforms or virtual reality therapy contribute to improving specific cognitive functions. The virtual realm also extends to support groups and educational webinars, fostering a sense of community for both patients and caregivers. Telehealth applications further aid in medication management, offering reminders and real-time monitoring. Counseling sessions and behavioral interventions can be conducted remotely, addressing emotional well-being and managing behavioral challenges associated with dementia. Additionally, telehealth facilitates care coordination through remote planning meetings, ensuring a comprehensive and coordinated approach to dementia care. The need arising from the COVID-19 pandemic catalyzed the rapid integration of tele-health systems, revolutionizing the landscape of sensory therapy interventions within the healthcare domain. Since conventional face-to-face therapy sessions faced constraints during the COVID-19 pandemic, the incorporation of virtual sensory therapy sessions emerged as a pivotal response, which employed advanced digital communication technologies to bridge the gap between therapists and patients [10,11,12]. Leveraging cutting-edge telecommunication infrastructure and sophisticated digital tools, therapists were able to effectively conduct real-time sensory assessments, interventions, and continuous monitoring, facilitating tailored treatment plans for patients. This assimilation of telehealth services extended the outreach of sensory therapy in establishing robust frameworks for data security, privacy, and regulatory compliance, ensuring the confidentiality of patient information and upholding ethical standards within the digital healthcare milieu [13,14].Integration of Mindfulness and Behavioral Strategies: Sensory diets are increasingly incorporating mindfulness techniques and behavioral strategies to help individuals to develop self-awareness and self-regulation skills. These strategies can complement sensory interventions, promoting long-term positive outcomes [15]. The key considerations for relevant outcome measures include sensory processing assessments such as the Sensory Profile or Sensory Processing Measure, facilitating the delineation of sensory preferences [16]. Functional outcomes can be evaluated through tools like the Functional Independence Measure for adults, providing granular insights into the impact on daily activities [17]. Assessing the broader impact on quality of life necessitates instruments like the Short Form Health Survey. In light of the increasing integration of mindfulness techniques, scales like the Mindful Attention Awareness Scale or the Self-Regulation Questionnaire become pertinent to quantitatively assess improvements in self-awareness and self-regulation skills [18].

As a result, the trajectory of sensory diets represents a promising avenue for addressing complex sensory processing challenges. With an emphasis on personalized methodologies, technological integration, and collaborative interdisciplinary efforts, it offers multifaceted solutions. Continued research and the integration of advanced technologies are poised to further optimize the efficacy and accessibility of sensory interventions, benefiting individuals navigating sensory processing difficulties.

Regarding current interventions for dementia (especially Alzheimer’s disease, which is the most common type of dementia), there are pharmacological and non-pharmacological interventions. In the case of elderly people with dementia, it is difficult to continue drug therapy over the long term due to age-related declines in liver and kidney function, the presence of comorbidities, drug-induced adverse events, and problems with medication adherence. In contrast, non-pharmacological interventions are relatively easy to implement as they do not have these problems and can be implemented regardless of the symptoms or stage of the disease. There are various types of non-pharmacological interventions for dementia (e.g., cognitive training, physical exercise, dietary treatments, art-oriented therapy, reminiscence therapy, and aromatherapy) [19,20,21,22]. Since the sensory diet can provide a tailored approach to modulating sensory experiences and triggering neuroplastic changes in the brain in individuals with dementia, we hypothesize that the sensory diet (especially when integrated with the latest technology) is one of the ideal non-pharmacological interventions for dementia.

This review aimed to investigate the current knowledge regarding the sensory diet and its potential application to dementia. 

## 2. Methods

To conduct a comprehensive review on sensory integration as a novel approach for healthy aging and dementia management, a systematic and rigorous methodology was employed. The search strategy aimed to identify relevant articles from key databases, and the selection criteria were defined to ensure the inclusion of high-quality and pertinent literature.

A thorough search was conducted across prominent databases, including PubMed and Google Scholar. These databases were chosen for their extensive coverage of the scientific literature across multiple disciplines, providing a well-rounded perspective on sensory integration in the context of healthy aging and dementia management.

This review included articles published within the timeframe of 2010 to 2023. This temporal range was selected to capture the most recent advancements and insights in the field of sensory integration, ensuring the inclusion of contemporary research that aligns with the review’s focus on novel approaches for healthy aging and dementia management.

The search strategy employed a combination of carefully selected keywords to identify the relevant literature. Keywords such as “sensory integration”, “healthy aging”, “dementia”, and “sensory modulation” were used to refine the search and target articles specifically addressing the intersection of sensory processing and the aging or dementia population.

The inclusion criteria encompassed peer-reviewed articles, systematic reviews, and meta-analyses. This broad range of materials was chosen to provide a comprehensive overview of the current state of research on sensory integration in healthy aging and dementia management. By including various types of materials, the review aimed to capture diverse perspectives and methodologies employed in the field.

The selected articles underwent a meticulous screening process based on predefined inclusion and exclusion criteria. The data extraction process involved synthesizing key findings related to the impact of sensory integration on cognitive function, emotional well-being, and overall quality of life in aging individuals and those with dementia.

Lastly, quantitative and qualitative data were analyzed to derive meaningful insights. Quantitative evaluations, such as sensory processing scales and advanced neuroimaging methodologies, were given particular attention to provide a robust evidence base for the effectiveness of sensory integration interventions.

## 3. Understanding Sensory Needs in Aging

The aging process involves an intricate mix of physiological changes, particularly affecting sensory perception. In the context of healthy aging, the varied weakening of people’s sensory modalities represents a natural progression that individuals undergo as they age. Aging brings about shifts in visual perception, affecting tasks like reading and recognizing small objects. The decline in visual sharpness, coupled with compromised motion perception, complicates spatial orientation. This is particularly challenging for patients, as they face difficulties in fine visual tasks, interpreting object motion, and judging distances. Therefore, personalized interventions, including environmental adjustments and technology use, are crucial to address the diverse aspects of age-related and neurodegenerative changes in visual perception [23,24,25]. 

Similarly, proprioception, which is the body’s intrinsic ability to sense its position and movement in space, is a fundamental concept relevant to individuals across the lifespan. In healthy individuals, proprioception underpins precise motor control, spatial awareness, and the maintenance of balance and posture. As individuals age, proprioception becomes a pivotal factor in addressing various challenges associated with aging. The decline in proprioceptive abilities leads to an increase in the risk of falls among older adults. Interventions targeting proprioception, including specific exercises and activities, play a crucial role in mitigating this risk, promoting joint health, and maintaining muscle tone. Proprioceptive training is integral to rehabilitation programs for seniors recovering from injuries or surgeries, fostering neuromuscular control and supporting overall mobility. Additionally, preserving proprioceptive function in aging individuals holds cognitive benefits which thus contribute to the integration of sensory information and potentially mitigating cognitive decline [26,27]. 

To address balance and proprioception within a sensory diet for healthy aging, various components play a crucial role; these include proprioceptive exercises, with balance training and weight shifting activities emerging as key contributors. Complementing this, the creation of multisensory environments has become integral, stimulating multiple senses simultaneously and fostering enhanced sensory integration. These environments may feature textured surfaces, varied lighting, and auditory cues, collectively contributing to improved proprioceptive feedback [28].

The inclusion of mind–body exercises like tai chi and yoga further enriches the sensory diet for healthy aging. Beyond promoting flexibility and strength, these practices emphasize body awareness, balance, and coordination, offering valuable elements to support overall proprioceptive well-being. Moreover, the consideration of footwear and supportive surfaces becomes paramount. Appropriate footwear and strategically designed home surfaces provide additional proprioceptive input, promoting stability and serving as preventive measures against the risk of falls [29].

Compromised depth perception is another facet of healthy aging which involves difficulties in accurately perceiving distances and spatial relationships [30]. This aspect becomes particularly relevant in activities where judging depth is crucial, such as navigating stairs, reaching for objects, or participating in sports. The aging process also brings about heightened sensitivity to glare, making bright lights and sunlight more uncomfortable and potentially disruptive to daily activities [31].

These sensory changes collectively present substantial challenges in performing fundamental tasks essential for daily living. For instance, due to various biological alternations in the signaling of neurons, activities like reading may require increased effort or the use of visual aids, driving can become more challenging due to altered depth perception and sensitivity to glare, and navigating unfamiliar environments may pose additional obstacles.

At the same time, in dementia conditions like Alzheimer’s disease, which cause a decline in thinking abilities, the brain faces challenges in adapting and forming new connections, a process known as neuroplasticity. Alzheimer’s disease causes insoluble deposits of beta-amyloid plaques and tau tangles that disrupt communication between brain cells. While the brain tries to compensate for cell loss, it can struggle to make enough changes, especially as the disease gets worse. Factors like education and staying mentally active, known as cognitive reserve, play a major role in how symptoms manifest themselves. Also, changes in brain chemicals, like not having enough acetylcholine, can contribute to a decline in thinking ability.

The interconnected nature of these sensory changes underscores the complexity of the challenges faced by the aging population. Comprehensive scientific approaches are imperative for understanding and addressing these multifaceted issues, offering avenues to enhance the quality of life for elderly individuals. Research into neuroplasticity, the underlying molecular mechanisms, and sensory system alterations has contributed to the development of targeted sensory diet interventions which aim at mitigating the impact of age-related sensory decline and cognitive impairment [32].

### The Role of Sensory Diet in Dementia Management

In the landscape of dementia management, the role of sensory diets stands as a compelling avenue of research, providing a tailored and integrative approach to alleviate symptoms and enhance the well-being of individuals affected by this complex neurological condition. While pharmacological interventions remain a cornerstone of treatment, the growing recognition of non-pharmacological approaches, such as sensory diets, is transforming the landscape of dementia care. Figure 1 provides a visual representation of the evolving health challenges that emerge during the aging process. This graphical depiction serves to highlight the complex nature of the health issues faced by individuals as they age. Similarly, this figure introduces and emphasizes the presence of tailored therapeutic solutions designed to address and mitigate these challenges. The visual presentation underscores the dynamic relationship between the aging process and the personalized interventions aimed at promoting health and well-being. 

Neuroplasticity, the capacity of the brain to reorganize itself through synaptic and structural modifications, plays a pivotal role in shaping cognitive trajectories across the human lifespan. In the developmental phase, particularly during early life, neuroplasticity is prominently evident, facilitating the acquisition of cognitive skills, memory formation, and the establishment of neural circuits critical for learning. As individuals progress through the aging continuum, the brain undergoes intricate changes in both the structure and function, as illustrated in Figure 2. Despite the inevitability of some decline, neuroplasticity remains a crucial mechanism for the adaptive recalibration of neural networks. 

In healthy aging, neuroplasticity plays a crucial role in mitigating cognitive decline. Individuals who are actively engaged in intellectually stimulating activities, maintain robust social connections, and pursue continuous learning endeavors exhibit a capacity to induce favorable neuroplastic changes. This phenomenon contributes to the concept of cognitive reserve, wherein the brain’s resilience to age-related alterations is fortified, leading to sustained cognitive abilities despite structural transformations.

Conversely, the narrative shifts markedly in the landscape of dementia-related aging. Here, neuroplasticity encounters formidable challenges. The accumulation of aberrant protein aggregates disrupts synaptic transmission and undermines the adaptive potential of neuroplasticity. Consequently, the brain’s ability to form new connections and adapt to changing circumstances is severely compromised, contributing significantly to the cognitive deterioration observed in dementia.

Comprehending the nuanced interplay of neuroplasticity in these diverse contexts is imperative for devising strategies to optimize cognitive health. In the realm of healthy aging, interventions aimed at promoting neuroplasticity-enriching activities, such as cognitive training and physical exercise, emerge as potential pathways to support cognitive well-being.

Fostering an understanding of the intricate interplay between sensory inputs and cognitive responses, the implementation of individualized sensory diets has gained traction for its potential. Tailoring interventions to meet specific sensory needs, healthcare professionals have demonstrated a profound commitment to enhancing the overall well-being and quality of life for individuals grappling with the complexities of this neurodegenerative condition [33].

The incorporation of sensory diets into dementia management offers a range of benefits primarily centered around their ability to mitigate the emotional distress and behavioral symptoms often associated with the condition [34]. Such interventions have been found to contribute to improving the sense of well-being and enhancing the overall quality of life of patients. For instance, in a recent cross-sectional study, Wójick et al. aimed to assess the current usage and acceptance of technology, specifically smartphones and computers, among 102 dementia caregivers, with a focus on socio-demographic factors. The findings of this study indicated that a significant majority of caregivers, particularly women, used smartphones and computers. This study further revealed that age, gender, and education level influenced the acceptance of technology, with smartphone use being more widespread across all age groups, whereas computer use was more common among younger caregivers. The respondents perceived technology as highly beneficial for facilitating various activities such as locomotion, toileting, and meal management for the patients [35]. Likewise, in a prospective investigation, Hwang et al. studied 2051 people and found that having problems with both seeing and hearing, known as dual sensory impairment, significantly increases the risk of all-cause dementia and Alzheimer’s disease. This study highlights the importance of addressing sensory issues, especially when both vision and hearing are affected, as potential factors contributing to dementia. The authors suggested that working on these sensory challenges could be a way to prevent dementia, and more research is needed in this area [36].

Furthermore, sensory diets have demonstrated effectiveness in regulating sleep patterns, which are commonly disrupted in individuals with dementia. Sleep disturbances not only exacerbate cognitive decline but also contribute to increased agitation and confusion. Evidence derived from recent studies highlights the efficacy of integrating sensory activities aimed at inducing relaxation and creating a conducive sleep environment for individuals with dementia [37]. These interventions have been associated with notable improvements in overall mood and cognitive functioning. Such positive outcomes emphasize the critical role of sensory-based interventions in optimizing the care and management of individuals living with dementia, pointing towards a promising avenue for enhancing their quality of life [38].

In addition to their impact on emotional well-being, sensory diets play a crucial role in preserving cognitive abilities and stimulating neural plasticity in individuals with dementia. By incorporating activities that engage various senses, such as reminiscence therapy, multisensory experiences, and cognitive games, these diets help to maintain cognitive function, including memory, attention, and executive function, as presented in Figure 3. Regular engagement in cognitive-stimulating sensory activities not only aids in the preservation of cognitive abilities but also promotes social interaction, reducing the feelings of isolation and loneliness that often accompany the progression of the disease.

Sensory diets also contribute significantly to the overall physical well-being of individuals in both healthy and dementia aging. By integrating sensory activities that involve movement, balance, and coordination, caregivers can help to maintain muscle strength, joint flexibility, and overall physical health. Engaging in activities such as light exercise, dance therapy, or sensory-based physical activities not only promotes physical health but also enhances the overall sense of well-being and self-confidence in individuals in both cases. Additionally, these activities help to reduce the risk of falls and injuries, which can be particularly detrimental to individuals with impaired cognitive function [39,40].

The positive impacts of sensory diets on cognitive function are closely linked to the fundamental brain mechanisms that regulate sensory processing, integration, and neuroplasticity. These diets, involving systematic exposure to diverse sensory experiences, stimulate neuroplasticity by influencing synaptic plasticity, thereby optimizing the efficiency of neuronal connections critical for learning and memory [41]. Leveraging the brain’s intricate sensory integration mechanisms, sensory diets promote the integration of diverse sensory inputs, enhancing the brain’s ability to create a cohesive perception of the environment and facilitating improved cognitive processing. The activation of specific brain regions in response to varied stimuli ensures holistic cognitive stimulation, while the release of neurotransmitters such as dopamine and serotonin contributes to mood regulation and attention [42]. Furthermore, sensory diets, tailored to individual preferences, finetune the reticular activating system, promoting sustained attention and focus. Importantly, these interventions also influence stress response systems, reducing cortisol levels and creating a conductive environment for optimal cognitive functioning. Thus, the positive impact of sensory diets on cognitive function arises from their ability to modulate neuroplasticity, promote sensory integration, activate specific brain regions, regulate neurotransmitter release, enhance attention, and alleviate stress [43]. Understanding these underlying brain mechanisms is crucial for designing effective sensory interventions that cater to individual needs and contribute to overall cognitive well-being.

## 4. Implementing a Customized Sensory Diet: Strategies for Effective Application

Implementing a customized sensory diet has gained traction as a non-pharmacological intervention that can significantly improve the quality of life for individuals. A sensory diet tailored to the specific needs and preferences of each individual can provide a structured and engaging routine that fosters emotional well-being, cognitive stimulation, and physical health. Different strategies for effectively implementing a customized sensory diet involve [44]: Comprehensive Assessment: The successful implementation of a sensory diet begins with a comprehensive assessment of the individual’s sensory preferences, cognitive abilities, and current emotional and physical well-being. Understanding the specific sensory sensitivities and aversions of the person with dementia is crucial for tailoring the sensory diet to their unique requirements.Individualized Plan Development: Based on the assessment, a personalized sensory diet plan is created that includes a variety of sensory activities targeting different senses. Activities that the individual enjoys and that promote relaxation, cognitive engagement, and physical activity are incorporated. The plan should be flexible and adaptable to accommodate changes in the individual’s condition over time. For instance, Rivan et al. conducted a prospective study among Malaysian community-dwelling older adults aimed at exploring the impact of dietary patterns on mild cognitive impairment and dementia incidence. The 5-year follow-up analysis of 280 participants aged 60 years and above encompassed various assessments, including cognitive, psychosocial, and functional evaluations, along with dietary intake data. The result of the study indicated that the “local snacks-fish and seafood-high salt foods” pattern increased the risk of mild cognitive impairment, while the “tropical fruits-oats” pattern showed a protective effect against dementia. These findings underscore the importance of dietary choices in influencing cognitive outcomes in the aging population [4].Multi-Sensory Stimulation: Integrating activities that engage multiple senses simultaneously helps to maximize the benefits of sensory stimulation. This may include incorporating music therapy, aromatherapy, tactile experiences, and visual stimuli in a coordinated manner to create a rich and immersive sensory environment that promotes emotional comfort and cognitive engagement [45], as presented in Figure 3. For instance, in a randomized controlled trial, Sánchez et al. examined the efficacy of a multisensory stimulation environment (MSSE) versus one-to-one activity sessions in managing severe dementia among elderly participants over 16 weeks. Significant improvements in neuropsychiatric symptoms and dementia severity were observed in the MSSE group compared to the one-to-one activity group. Both cohorts experienced reduced agitation, with no statistically significant distinctions. The study indicates that MSSE may offer heightened effectiveness in addressing severe dementia symptoms compared to one-to-one activities [46].Regular Evaluation and Adjustment: It is important to regularly evaluate the effectiveness of the sensory diet in meeting the individual’s emotional, cognitive, and physical needs. The plan can be adjusted as necessary to accommodate changes in the individual’s preferences, abilities, and overall condition. Continual monitoring and adaptation are essential to ensure that the sensory diet remains relevant and beneficial [5].

## 5. Innovative Convergence: Exploring the Intersection of Sensory Diets and Technology for Enhanced Support in Sensory Processing Difficulties

In recent years, the convergence of sensory diets and technology has ushered in innovative solutions to support individuals in managing their sensory needs. The amalgamation of different fields resulted in a range of applications and devices aiming to provide tailored sensory experiences to aid individuals in their daily lives, as presented in Figure 4.

The widespread availability of sensory apps stands out as a significant development. These apps are tailored to address the complexities of the human sensory system, encompassing visual, auditory, tactile, gustatory, and olfactory modalities. Each modality contributes uniquely to our holistic sensory experience, shaping how we perceive and engage with the dynamic world around us. In the realm of personalized sensory diets, the acknowledgment and incorporation of these diverse modalities become paramount. A comprehensive sensory diet strives to incorporate a variety of activities and stimuli, catering to an individual’s distinct sensory preferences and sensitivities. Whether it involves engaging visual senses with vibrant colors or stimulating tactile senses through diverse textures, the focus on diverse modalities ensures a well-rounded and tailored approach to sensory stimulation within the framework of a sensory diet. Additionally, the advent of sensory-related apps introduces a technological dimension to this discourse, with evaluations using tools like the Mobile App Rating Scale (MARS) providing insights into the efficacy of these apps in meeting diverse sensory needs. The integration of insights from different modalities and app evaluations underscores the intricate relationship between technology and sensory well-being, offering a comprehensive perspective on how these advancements contribute to our understanding and enhancement of the sensory experience. 

As a result, it can be said that these apps play a crucial role in assisting individuals with sensory processing difficulties, thus offering a range of activities crafted to provide targeted sensory input. Whether through calming sounds that promote relaxation or visually engaging graphics encouraging focus and participation, these apps serve as invaluable tools for individuals seeking to regulate and improve their sensory experiences. The interactive nature of these apps further fosters engagement and active participation, contributing to sustained improvement in sensory regulation over time [47]. 

Expanding the discourse beyond the digital realm, the incorporation of technology into wearable sensory solutions assumes heightened significance within the scientific framework, particularly concerning the complexities of sensory aging. Wearable devices, endowed with specialized mechanisms, manifest as pragmatic tools, offering a tailored approach for dispensing pressure input and sensory stimulation. This innovation is intricately designed to furnish a continuous and personalized source of sensory input throughout the day, thereby assuming a pivotal role in supporting individuals in their sensory regulation endeavors throughout the aging process.

The seamless integration of technology into these wearables not only underscores their efficacy but also emphasizes a commitment to ensuring user comfort and convenience within the scientific context. By discreetly amalgamating technology, these wearables provide individuals with an unobtrusive means of accessing crucial sensory support, preserving the integrity of their sensory experiences while navigating daily activities without undue interference.

In the specific domain of sensory aging, this integration holds substantial promise as a scientifically grounded and effective avenue for addressing the nuanced sensory needs within this demographic. The seamless fusion of technology into wearables enables a refined and personalized approach to sensory regulation, acknowledging the intricate sensory profiles and evolving requirements of individuals as they age. This approach not only signifies technological advancement but also underscores a compassionate understanding of the challenges associated with sensory aging. It reflects a commitment to advancing scientific knowledge while enhancing the overall well-being and quality of life for individuals within this demographic [48].

Additionally, the term “metaverse” denotes a virtual shared space that extends beyond the physical realm, typically accessed through the Internet. It integrates VR, AR, and other immersive technologies to form a continuous and interconnected digital universe. Distinguishing itself from VR and AR, the metaverse aspires to create a seamless and enduring online environment, promoting real-time interaction among users and digital content.

The metaverse boasts several advantages over VR and AR, particularly in its interconnected design, allowing users to effortlessly navigate diverse virtual environments and experiences. This interconnectedness fosters a more cohesive and immersive digital world, enabling a broader spectrum of activities and interactions.

In the context of supporting individuals with cognitive impairments, such as dementia, the metaverse holds potential for enhancing cognitive abilities and overall well-being. Within the metaverse, immersive virtual environments can replicate familiar real-life settings and activities, offering therapeutic experiences that contribute to memory consolidation, cognitive stimulation, and emotional regulation. This surpasses the capabilities of traditional VR and AR, as the metaverse allows for a more extensive and interconnected array of experiences.

For those with dementia, VR simulations within the metaverse can faithfully recreate past experiences, supporting reminiscence therapy. Furthermore, the metaverse can offer tailored interactive cognitive exercises to suit individual cognitive profiles. Crucially, it acts as a conducive platform for fostering social interaction, addressing the common issue of social isolation experienced by individuals with cognitive impairments. By facilitating virtual social connections, the metaverse plays a role in enhancing overall mental well-being.

Despite these benefits, it is worth noting that the advent of VR and AR has already opened up possibilities for creating controlled sensory environments. Through these technologies, individuals can immerse themselves in simulated settings with customizable sensory experiences, tailored to specific preferences for a secure and controlled engagement. Leveraging VR and AR provides access to a diverse range of sensory stimuli that can be adjusted to meet individual sensory requirements [49,50].

Nonetheless, the ethical implications, privacy concerns, and accessibility issues must be carefully considered and addressed to ensure responsible and effective integration of these technologies in supporting individuals with cognitive limitations [51,52].

Tech companies have also responded to the demand for sensory-friendly devices by introducing specialized product lines that cater to individuals with sensory sensitivities. For instance, companies have focused on developing sensory-friendly features for smartphones and tablets. Products such as iPhones and iPads from Apple come equipped with options for reduced screen brightness, customizable color filters, and integrated noise-cancelling features. Meanwhile, Samsung, which is a major player in the tech industry, has incorporated similar sensory-friendly features into its smartphones and tablets, offering users the ability to adjust screen brightness, apply customizable color filters, and access built-in noise-cancelling options. These adaptations aim to create a sensory-friendly user experience, particularly beneficial for individuals who may be sensitive to bright lights, specific color contrasts, or excessive ambient noise. By incorporating these sensory-friendly features, technology companies are fostering inclusivity and accessibility, ensuring that individuals with sensory challenges can seamlessly integrate technology into their daily lives.

As the intersection of sensory diets and technology continues to evolve, the ongoing innovations and integration of these solutions hold significant promise in enhancing the quality of life for individuals with sensory processing difficulties. 

## 6. Conclusions

The sensory diet paradigm represents a scientifically grounded and pragmatic approach for optimizing healthy aging and mitigating the challenges associated with dementia. Anchored in the profound influence of multisensory experiences on neurocognitive and emotional parameters, the sensory diet framework provides a nuanced, individualized strategy to elevate the overall quality of life in aging populations and those grappling with dementia. Through the systematic integration of diverse sensory stimuli encompassing tactile, auditory, visual, and olfactory modalities, this approach transcends conventional care models. A closer examination reveals a potential emphasis on two primary modalities—tactile and visual. While recognizing the undeniable significance of tactile and visual stimuli in fostering cognitive enhancement and overall well-being, it is imperative to acknowledge the need for a more inclusive representation of auditory and olfactory modalities.

Within the rich tapestry of sensory experiences, tactile stimuli offer tangible connections with the physical environment, providing a foundation for engagement and interaction. Visual stimuli, on the other hand, open windows to perception, creating immersive environments that facilitate cognitive stimulation. Despite their undeniable importance, an exploration of auditory modalities introduces an additional layer of complexity, as sound can evoke emotional responses, trigger memory recall, and provide a dynamic dimension to the sensory landscape.

Moreover, an emphasis on olfactory modalities further enriches the sensory narrative. Scents have the profound ability to evoke emotions, unlock memories, and influence cognitive states. The integration of aromatherapy techniques and carefully selected scents becomes pivotal in creating a holistic sensory environment, addressing not only cognitive aspects but also emotional well-being.

In essence, while tactile and visual modalities form the cornerstone of the sensory integration approach, the true innovation lies in the harmonious orchestration of diverse sensory elements. By extending the exploration to auditory and olfactory realms, this approach transcends conventional boundaries, offering a more profound and nuanced understanding of how multi-modal sensory integration contributes to a holistic and enriched care paradigm. This holistic perspective aligns seamlessly with the overarching goal of creating a sensory environment that is not just therapeutic but also reflective of the complexity and richness of human experience.

In navigating the intricate facets of the aging process, the adoption of sensory diet principles not only signifies an innovative scientific pursuit but also underscores a compassionate and empirically validated avenue to cultivate resilience, uphold dignity, and instill a profound sense of purpose throughout the continuum of aging.

## Figures and Tables

**Figure 1 brainsci-14-00285-f001:**
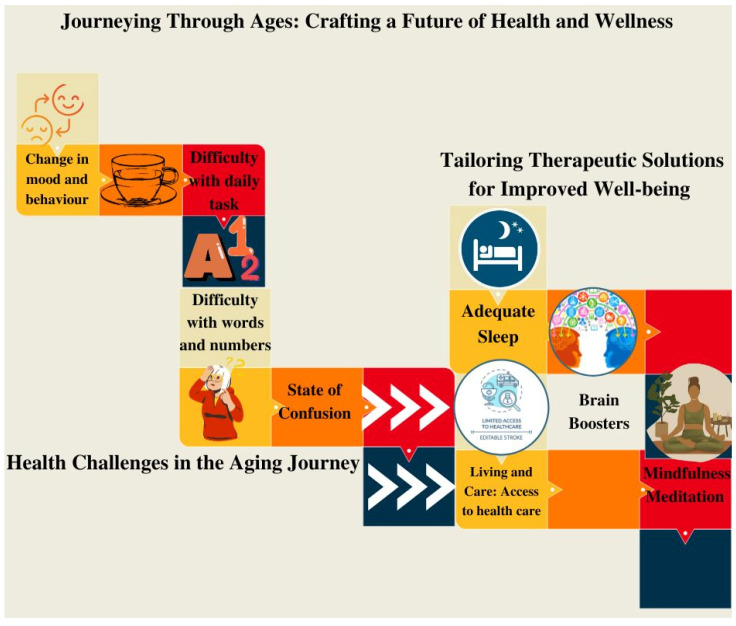
Schematic representation reflecting health challenges during aging and the tailored therapeutic solutions.

**Figure 2 brainsci-14-00285-f002:**
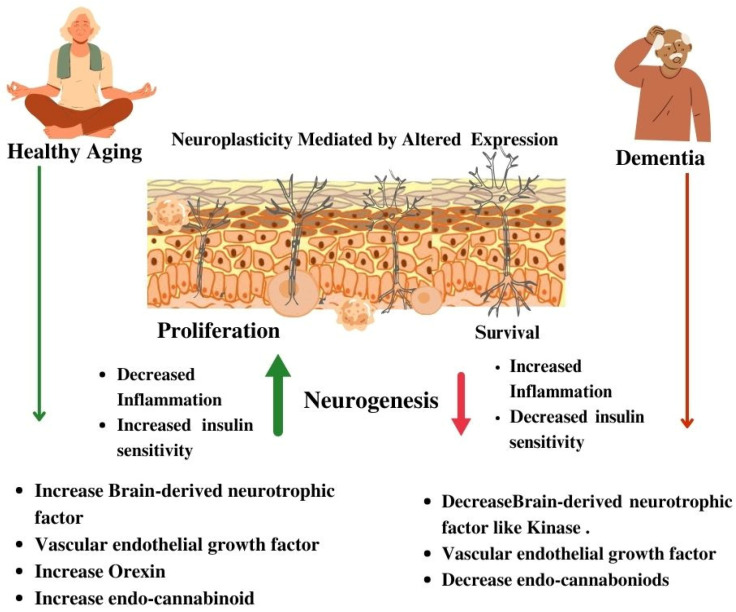
Schematic representation reflecting neuroplasticity changes.

**Figure 3 brainsci-14-00285-f003:**
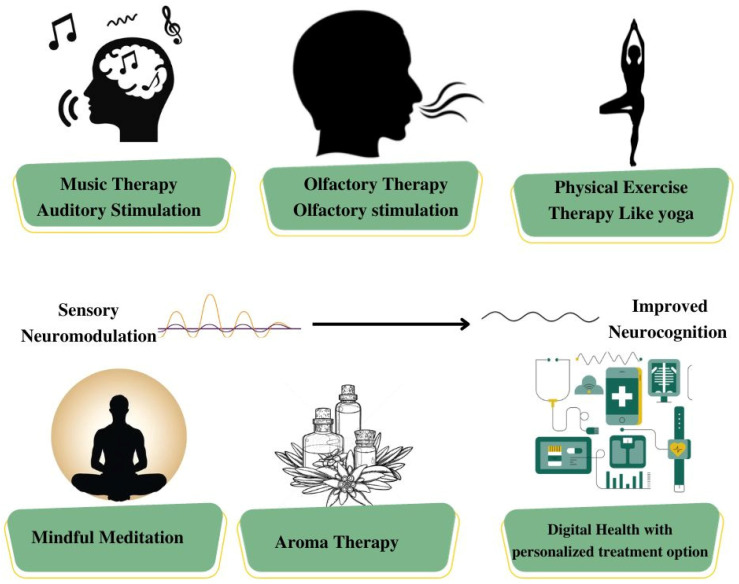
Schematic representation of different sensory activities for healthy patients and the well-being of dementia patients.

**Figure 4 brainsci-14-00285-f004:**
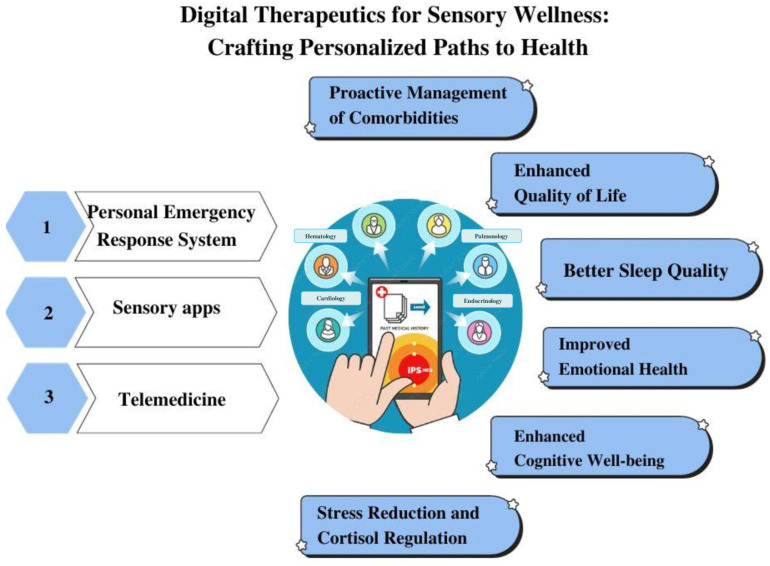
Schematic representation of technological solutions for different aging communities.

## Data Availability

Not applicable.

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
