# Peer review of "Sensory Integration: A Novel Approach for Healthy Ageing and Dementia Management"

_brainsci, 2024, doi:10.3390/brainsci14030285_

Round 1

Reviewer 1 Report

Comments and Suggestions for Authors

Kindly refer to the attached document. I suggest to deepen the depth of the scientific treatment of the content. Current content seems broad and too generic, e.g. "based on neural plastic changes", "VR technology is beneficial", but the authors shall share more and discuss using reference to prior studies.

Author Response

To Reviewer 1:

Comment 1: The structure of the manuscript is alright. However, the depth of the review is not up to expectations. Instead, a first glance will tell readers in the field that the content is more or less like a textbook summary.

Response: Thank you for your valuable suggestion. In this revised version, we have extensively revised the entire manuscript.

Comment 2: The content lacks comparative and argumentative aspects. For example, they should discuss more on why ‘sensory diet’ is recommended by them as opposed to other conventional management tools. Thus, current gaps shall be clearly introduced in the Introduction.

Response: Thank you for pointing out an important point. We have added discussion regarding the possibility of recommending a ‘sensory diet’ over other interventions (Lines 132-145).

Comment 3: Similarly, the concept of sensory diet itself is not ‘novel’. It is often employed in child development where the neuroplastic changes are yet to reach an optimal level. Dementia is considered challenging for therapists and clinicians alike.

Response: We understand that the concept of ‘sensory integration’ is not new. In this revised version, we have described in more detail the novelty of integrating advanced technology with traditional ‘sensory diet’ and applying them to dementia (Lines 65-150).

Comment 4: The abstract is correctly structured, but the background (first few sentences) is too long before the introduction of the study objectives. You may move the objectives earlier up.

 Response: We have extensively rewritten the Abstract section (Lines 18-40). The first few sentences have been moved to the Introduction section (Lines 44-50).

Comment 5: The objectives said that the manuscript discusses the effectiveness of the sensory diets. I suggest the authors be more focused and end the abstract with some conclusive statement about what the authors have found, rather than creating a vague recommendation. For example, they can write how many sensory modalities are included in the review (e.g. visual, auditory) and the types of therapy which are proven to show some efficacy, etc.

Response: According to your comments (comments 4 and 5), we have extensively rewritten the Abstract section (Lines 18-40).

Comment 6: The content lacks depth when mentioning past studies. The authors shall add what the study findings are when citing them, whether the study cited is an cross-sectional or RCT, and so on.

Response: In accordance with your comments, we have tried to include as much detail as possible about past studies in this revised version.

Comment 7: I invite the authors to take a look at the following paper for reference. https://www.ncbi.nlm.nih.gov/pmc/articles/PMC1124787/

Response: We have added a discussion citing the paper you recommended (Lines 139-142).

Comment 8: Most sensory modalities mentioned in the manuscript are about vision. But sensory dysfunction in ageing also affects balance (proprioception), which gets degraded with age and increases fall risk in older adults. In the context of healthy ageing, please briefly discuss how ‘sensory diet’ possibly manages this.

Response: Thank you for your constructive comments. We have added a discussion on proprioception (Lines 195-221).

Comment 9: Comprehensive outcome measures for personalized intervention. Can the authors mention what type of relevant outcome measures? Include the appropriate citations too.

Response: We have added information about them (Lines 116-125).

Comment 10: The authors mentioned tech companies producing relevant products. Can the authors be more specific as to what product lines, for example, and whether past studies have included those in their work.

Response: We are grateful for your valuable suggestion. As per suggestion the products have been specifically defined (Lines 515-530).

Comment 11: The manuscript content is suitable for the journal.

Response: Thank you very much for your very valuable comments. We believe that the revisions made in accordance with your comments will make our manuscript even more suitable for journal publication.

Comment 12: The references follow the style recommended by the journal.

Response: According to the journal’s style, we changed the style of References (Lines 582-691).

Reviewer 2 Report

Comments and Suggestions for Authors

1. The review article is timely, as the global population is aging.

2. The abstract should highlight the review's aim and the methods followed for data collection. List the databases searched for the articles, the span of years covered, the keywords searched and the types of materials retrieved. 

3. The Methods section should be included in the body of the review article. 

Comments on the Quality of English Language

Moderate English revision is required. 

Author Response

To Reviewer 2:

Comment 1: The review article is timely, as the global population is aging.

Response: We sincerely thank you for evaluating our manuscript. We have revised the manuscript according to your comments. We believe that this revision made our manuscript more suitable for the journal.

Comment 2: The abstract should highlight the review's aim and the methods followed for data collection. List the databases searched for the articles, the span of years covered, the keywords searched and the types of materials retrieved.

Response: We followed your comments and highlighted those information in the Abstract section (Lines 21-27).

Comment 3: The Methods section should be included in the body of the review article.

Response: Following your comments, we added the Methods section (Lines 151-183).

Comment 4: Moderate English revision is required.

Response: We have rectified the quality of English thoroughly in the manuscript. 

Reviewer 3 Report

Comments and Suggestions for Authors

Thank you for the opportunity to comment on this review.

The review concerns the sensory diet as an approach to dementia management and healthy aging. While informative and broad in the included scope, the review is presented in a free flowing style and lacks the level of preciseness in description of key concepts. I highlight these below:

  • The authors do not present discuss sensory input in detail in the initial part of the manuscript. Further, even within the sesory modality of vision, the authors are not clear on what they are discussing. For instance, on P3, the authors loftily describes "alterations in sensory perception" and a "multifaceted decline in sensory modalities represents a natural progression that individuals undergo as they age". In the paragraph following, the authors describe specifically depth perception. In fact, the first three paragraphs can likely be condensed to a much clearer presentation. 
  • L72-85 Telehealth application of interventions for dementia did not occur as a result of COVID-19. Please provide a more accurate description of dementia telehealth interventions.
  • In contrast with the vague descriptions of sensory inputs, and on L244 "Brain regions", the authors proceed to discuss two examples of neurotransmitters (L245) and specific stress hormones (L248) involved. L286-294 the authors also provide a very detailed account of a single study, and I cannot understand why this was considered motivated given the level of detail used in the remaining test. Please reconsider the level of details given.
  • On P9 (L310 and onwards) the authors present the idea that there are apps that provide sensory input. I would suggest that the authors should also include a reflection on the number of modalities in which this sensory inputs are given. And please relate this discussion to the sensory diet. Please also highlight the added benefit of including the discussion of MARS scores of apps.
  • On L342 the authors introduce the metaverse, after having discussed Virtual and Augmented Realities in the preceding paragraph . Please provide a definition and description of the metaverse and expand on the added benefit that this concept over the VR and AR discussion already provided.
  • On L371 the authors conclude that the integration of sensory input from a diverse set of modalities is an approach that transcends conventional care models. However, one may argue that sensory input is given primarily in two sensory modalities in the stimulation strategies that the authors describe in detail. Please expand throughout the manuscript.

Minor points

  • L53 Please reconsider the use of repertoire in this context.
  • Please revise all sentences to avoid repetitions. Refer to L61-64 where "sensory" is used three times, and two times within the same sentence. Similar patterns of excessive repetition of words are  in, for instance, L68-70. Please revise the text for the benefit of the reader.
  • The purpose or importance of Figure 1 needs to be made substantially clearer in the manuscript.
  •  The link between the text on L144-150 and Figure 2 needs to be substantially strengthened.
  • L202 In this sentence, the authors discuss sensory impairment, and then give dual sensory impairment as a particularly strong risk factor. But dual sensory impairment is no more specific than the very vague sensory impairment is already, so please provide a specification of which senses have actually been studied.
  • The purpose of Figure 3 is not clear, and the figure contains various levels of information. The distinction between the two parts of the figure (the top part and the bottom part) requires a description, as does the purpose of the TV set in the figure. What is the purpose of the figure in relation to sensory diet? How is the relation between a sensory diet and sensory overloading?
Comments on the Quality of English Language

The style of writing is sometimes a bit awkward for a scientific setting, but it can be easily amended.

Author Response

To Reviewer 3:

Comment 1: The authors do not present discuss sensory input in detail in the initial part of the manuscript. Further, even within the sensory modality of vision, the authors are not clear on what they are discussing. For instance, on P3, the authors loftily describes "alterations in sensory perception" and a "multifaceted decline in sensory modalities represents a natural progression that individuals undergo as they age". In the paragraph following, the authors describe specifically depth perception. In fact, the first three paragraphs can likely be condensed to a much clearer presentation.

Response: Thank you for your constructive comments. In this revised version, we have extensively added discussion of sensory input and tried to keep the presentation clear  (Lines 185-221).

Comment 2: L72-85 Telehealth application of interventions for dementia did not occur as a result of COVID-19. Please provide a more accurate description of dementia telehealth interventions.

Response: We have descried this matter correctly (Lines 84-112).

Comment 3: In contrast with the vague descriptions of sensory inputs, and on L244 "Brain regions", the authors proceed to discuss two examples of neurotransmitters (L245) and specific stress hormones (L248) involved. L286-294 the authors also provide a very detailed account of a single study, and I cannot understand why this was considered motivated given the level of detail used in the remaining test. Please reconsider the level of details given.

Response: We reconsidered these points and extensively revised the manuscript.

Comment 4: On P9 (L310 and onwards) the authors present the idea that there are apps that provide sensory input. I would suggest that the authors should also include a reflection on the number of modalities in which this sensory input are given. And please relate this discussion to the sensory diet. Please also highlight the added benefit of including the discussion of MARS scores of apps.

Response: We have discussed these points (Lines 424-448).

Comment 5: On L342 the authors introduce the metaverse, after having discussed Virtual and Augmented Realities in the preceding paragraph. Please provide a definition and description of the metaverse and expand on the added benefit that this concept over the VR and AR discussion already provided.

Response: Following your comments, we have added notes regarding these points (Lines 472-514).

Comment 6: On L371 the authors conclude that the integration of sensory input from a diverse set of modalities is an approach that transcends conventional care models. However, one may argue that sensory input is given primarily in two sensory modalities in the stimulation strategies that the authors describe in detail. Please expand throughout the manuscript.

Response: Following your comments, we have extensively revised the manuscript (Lines 542-565)

Comment 7: L53 Please reconsider the use of repertoire in this context.

Response: We have reconsidered this point (Lines 60-61).

Comment 8: Please revise all sentences to avoid repetitions. Refer to L61-64 where "sensory" is used three times, and two times within the same sentence. Similar patterns of excessive repetition of words are in, for instance, L68-70. Please revise the text for the benefit of the reader.

Response: We have tried as much as possible to avoid repetitions.

Comment 9: The purpose or importance of Figure 1 needs to be made substantially clearer in the manuscript.

Response: We have clarified the purpose and importance of Figure 2 in the revised version (this figure was Figure 1 in the original version (Lines 267-273).

Comment 10: The link between the text on L144-150 and Figure 2 needs to be substantially strengthened.

Response: Figure 2 in the original version is re-cited as Figure 1 in the revised version. A strong connection has been made making the conceptualization of the representation of the figure clear to the reader (Lines 256-262)

Comment 11: L202 In this sentence, the authors discuss sensory impairment, and then give dual sensory impairment as a particularly strong risk factor. But dual sensory impairment is no more specific than the very vague sensory impairment is already, so please provide a specification of which senses have actually been studied.

Response: We have changed the information regarding this matter to make it clear (Lines 314-320).

Comment 12: The purpose of Figure 3 is not clear, and the figure contains various levels of information. The distinction between the two parts of the figure (the top part and the bottom part) requires a description, as does the purpose of the TV set in the figure. What is the purpose of the figure in relation to sensory diet? How is the relation between a sensory diet and sensory overloading?

Response: Figure 3 in the original version has been removed to avoid confusion for readers in the revised version.

Comment 13: The style of writing is sometimes a bit awkward for a scientific setting, but it can be easily amended.

Response: We have rectified the quality of English thoroughly in the manuscript.

Round 2

Reviewer 1 Report

Comments and Suggestions for Authors

In the Introduction, the authors mentioned the drawbacks of some of the pharmacological interventions for dementia. Please provide 1-2 citations:

https://www.frontiersin.org/journals/pharmacology/articles/10.3389/fphar.2020.01168/full

This is important given that sensory diet is a form of behavioural, rather than pharmacological, type of intervention.

The authors have addressed most of the concerns.

Author Response

To Reviewer 1:

Comment 1: In the Introduction, the authors mentioned the drawbacks of some of the pharmacological interventions for dementia. Please provide 1-2 citations: https://www.frontiersin.org/journals/pharmacology/articles/10.3389/fphar.2020.01168/full This is important given that sensory diet is a form of behavioural, rather than pharmacological, type of intervention.

Response: Thank you for your suggestion. We have added two new citations (Line 142).

Reviewer 3 Report

Comments and Suggestions for Authors

The manuscript reads well and is in my view mostly acceptable for publication. 

I suggest that the authors remove the last paragraph in P3 (L132-145). I see why it it may seem that a summary of pharmacological treatments is warranted in any intervention review, the contexts in which sensory diets may be applied is vastly different, and the paragraph is therefore not needed.

Similarly, I suggest that the line on L146 can just begin with “This review..”  and contain just the first sentence. The effort to motivate the review further is superfluous.

- There is an extra ‘-‘ on L100

Author Response

To Reviewer 3:

Comment 1: I suggest that the authors remove the last paragraph in P3 (L132-145). I see why it it may seem that a summary of pharmacological treatments is warranted in any intervention review, the contexts in which sensory diets may be applied is vastly different, and the paragraph is therefore not needed.

Response: Thank you for your suggestion. However, this paragraph was added following a suggestion from Reviewer 1. Reviewer 1 also provided new suggestions regarding this paragraph in this revised version. Therefore, we would like to retain this paragraph.

Comment 2: Similarly, I suggest that the line on L146 can just begin with “This review..”  and contain just the first sentence. The effort to motivate the review further is superfluous.

Response: According to your suggestions, we have made changes (Lines 146-147).

Comment 3: - There is an extra ‘-‘ on L100

Response: We removed the extra ‘-‘ (Line 100).